# An Updated List of Rock Partridge (*Alectoris graeca*) Haplotypes from the Apennines—Central Italy

**Leonardo Brustenga** [1], **Paolo Viola** [2], **Pedro Girotti** [2], **Andrea Amici** [2], **Alessandro Rossetti** [3], **Stefania Chiesa** [4], **Riccardo Primi** [2,*], **Luigi Esposito** [5] and **Livia Lucentini** [1]

1 Department of Chemistry, Biology and Biotechnology, University of Perugia, 06123 Perugia, Italy
2 Department of Agriculture and Forestry Science, DAFNE, University of Tuscia, 01100 Viterbo, Italy
3 Sibillini Mountains National Park, 62039 Visso, Italy
4 ISPRA—The Italian Institute for Environmental Protection and Research, 00144 Rome, Italy
5 Department of Veterinary Medicine and Animal Production, University of Naples Federico II, 80137 Naples, Italy
* Correspondence: primi@unitus.it

**Abstract:** We report an updated and expanded list of Rock Partridge (*Alectoris graeca*) haplotypes found in wild animals throughout the Apennines of central Italy. Samples were collected and identified during a monitoring program of autochthonous Galliformes and from a private collection. The haplotypes were identified on a longer fragment of the mitochondrial control region (D-loop) based on previously reported haplotypes. This novel evidence, based on a wider sampling area and a higher number of analyzed specimens, will be of relevance in both conservation projects and gamebird breeding for restock, as imposed by the Italian Action Plan. Studying longer fragments can also be useful for phylogeographic analysis.

**Dataset:** Molecular data on the NCBI web site (GenBank) are available at the link https://www.ncbi.nlm.nih.gov. Registration codes are ON086324, ON086325, ON086326, ON086327, ON086328, ON086329, ON086317, ON086318, ON086319, ON086320, ON086322, ON086323.

**Keywords:** Rock Partridge; *Alectoris graeca*; haplotypes; D-loop; Apennines; central Italy; ex situ conservation

## 1. Summary

The Rock partridge (*Alectoris graeca* Meisner, 1804) is a stocky medium size *Phasianidae* Horsefield, 1821, characterized by blue-grey upperparts, a frontal white band above a black one that starts over the beak, which encompasses the eyes and merges on the chest as a collar delimiting a white chin and throat. The species also has white flanks feathers vertically barred by black and chestnut thin bands. Lore color is black and ear-covert colors is black with yellow extremities [1]. The species, endemic to Europe, is distributed in the Alps (France, Italy, Austria, Slovenia and Switzerland), in the Apennines (Italy), in Sicily (Italy) and in the Balkan Peninsula with sporadic occurrence in Romania and in Bulgaria, corresponding to a natural contact zone with the congeneric *A. chukar* [1]. Four subspecies, among which the Apennine one (*A. g. orlandoi*), were previously described principally on the base of plumage colors variations [2]. However, subsequent analysis based on a mitochondrial marker (D-loop) did not confirm this subspecific differentiation, highlighting, in particular, close affinity between the Apennine and the Balkan populations [3]. In Italy, the Apennine population is isolated and demographically independent from all the others, and therefore, it should be considered as a distinct management unit (MU) and an independent evolutionary significant unit (ESU) [4,5]. The Rock partridge was recently

listed in category 1 among the Species of European Conservation Concern (SPEC) [6], and the Italian population has been classified as Vulnerable [7,8]. The causes of these negative trend are mostly anthropogenic, as previously reported in several studies [9–13]. In particular, the Apennine Rock partridge, as well as the Sicilian one (*A. g. whitakeri*) [14], is raising concerns because it persists with small, scattered and isolated nuclei, and local extinctions are reported in all pre- and anti-Apennine sectors [5,15]. Despite this, it continues to be exploited in some regions in compliance with conservative hunting plans. To preserve residual populations, a rapid advancement of knowledge is necessary to guide conservation efforts towards the enhancing of wild populations, as advised by the Council of Europe [16]. Accordingly, Italy drew up and adopted a specific national action plan for Rock partridges' conservation [5] and the interest in monitoring, in situ management and ex situ conservation grew [17]. The plan imposes a priority phase of genetic characterization at population level to ensure the conservation of native biodiversity and local genotypes. The first Dloop-based phylogenetic tree showing the affinity between Apennine and Balkan *A. graeca* samples was established in 1998 [3]. Nevertheless, nineteen years ago, Randi et al. [4] provided an updated list of European haplotypes; the ones belonging to the Apennines were described from 37 specimens collected in 6 sampling areas [4], confirming the genetic affinity between Apennine and Balkan populations belonging to the nominal subspecies (*A. g. graeca*), previously described [3,4]. However, Randi et al. [4] declared unsolved and unsatisfactory the assessment of genetic diversification between populations from Alps (*A. g. saxatilis*), Apennines and Balkans, due to the small sample size and the lack of specimens from critical geographical areas. The present paper aims to report an updated and expanded dataset of Rock partridge haplotypes, improving the size and the distribution of the Apennine samples. Furthermore, the data reported allow the identification of previously undetected haplotypes and the description of new ones based on mtDNA fragments longer than the previously sequenced. These advancement in knowledge can support both ex situ conservation purposes addressing the selection of genetically suitable founders and a better understanding of the species' phylogeography, sharing long sequences from different crucial areas of the European distribution range.

## 2. Methods

### 2.1. Sampling

Aiming to increase both numerically and geographically the Apennine sample pool, biological samples from 69 specimens of Rock partridges (*Alectoris graeca*) (20 sampling areas, 59 sampling events) of the Central Apennine were collected. Non-invasive sampling was performed on 61 of the samples by collecting feathers and/or feces during an authorized monitoring program. The remaining 8 samples, dated late 1990s, were taxidermized, and sourced from a private collection. In the last case, feathers were drawn from different body parts considering the importance of minimizing the visible damage to the specimens. All biological samples (feces or feathers) were immediately placed in a test-tube and stored at +4 °C.

### 2.2. Procedures

DNA extraction as well as PCR amplifications were conducted following already validated protocols [13,18–20]. DNA extraction from feces is a largely standardized technique, the applicability of which is based on the feces conservation status. In fact, due to the possible presence of degrading enzyme or substances interfering with DNA extraction, the sampling campaign should be performed, allowing the refrigeration of samples as soon as possible. For these reasons, extraction from feathers is preferable, as the DNA is usually of better quality and quantity. Amplifications were made using two sets of primers to allow a Semi-Nested PCR re-amplification to enhance the DNA extracted from conservative samples. The first amplification was performed using PHDL and H1321 primers [18], the amplified products were purified and then two independent Semi-Nested PCRs were performed using SEMD621 and SEMD467 primers paired with PHDL and

H1321, respectively, with the aim to produce two overlapping fragments (A, B) [18]. Upon verification on 2.5% agarose gel, amplified products were purified with ExoSAP-IT$^{TM}$ (ThermoFisher Scientific, Waltham, MA, USA) according to manufacturer's instructions and later outsourced to Eurofins Genomic (eurofinsgenomics.eu) for sequencing. The fragment of the D-loop control region originally analyzed by Randi et al. [4] was doubled in length (from 430 bp to 896 bp) and screened for ulterior polymorphic sites. The obtained sequences were manually aligned with MEGA X and visually screened for polymorphisms. NUMTs (Nuclear mitochondrial DNA) were reasonably excluded considering fragment size and sequence identities, following the criteria already suggested [11,13].

### 2.3. Haplotypes' Attribution

All sequences were first aligned with haplotypes reported by Randi et al. [2], bestowing each sequence to one of the 430 bp haplotypes already described by these authors. Of the 69 analyzed samples, 8 contemporary samples were excluded, as they showed *A. chukar* matrilinear lineage (12%). Indeed, illegal translocations of Chukars or hybrids were widely performed, in both the Alps and Apennines, in the period between the mid-20th century and the beginning of the 21th at least [21]. Unofficial reports suggest the occurrence of recent and actual introduction in various mountain areas. The remaining 61 samples were matched to four haplotypes (H3, H8, H24 and H10) previously described by Randi et al. [4]. The mutations in the interval between position 431 and position 896 allowed for the identification of 7 new Apennine haplotypes, based on longer sequences. These haplotypes were named and registered with the nomenclature used by Randi et al. [2] followed by an alphabet letter and by "-long" for as many new haplotypes found, e.g., H3a-long, H3b-long and so on.

### 3. Data Description

In Table 1, we provide, for each wild Apennine sample (column one–Code), species' attribution (column two), type of sample (column three–Specimen), dating (column four), geographical origin (columns five and six–Province and Mountain area), haplotypes attribution on the base of the 430 bp sequences previously described by Randi et al. [4] (column seven) and haplotypes attribution on the base of the 896 bp sequences (column eight).

**Table 1.** Data from each genotyped sample of *Alectoris graeca*. In the second-to-last column the matching of the samples to the haplotypes found by Randi et al. [4] are reported, whereas the last column reports the matching to the 896 bp haplotypes described in this study. The length of the produced sequence is given whenever the 896 bp fragment could not be successfully sequenced in its entirety, preventing the long haplotype matching (e.g., sample Wild-8). Province: AP, Ascoli-Piceno; AQ, Aquila; FR, Frosinone; IS, Isernia; MC, Macerata; PG, Perugia; RI, Rieti.

| Code | Species | Specimen | Dating | Province | Mountain/Area | 430 bp-H | 896 bp-H |
|------|---------|----------|--------|----------|---------------|----------|----------|
| Wild-1 | *A. graeca* | Feathers | Contemporary | AQ | Calvo | H8 | H8a |
| Wild-2 | *A. chukar* | Feathers | Contemporary | RI | Duchessa | - | - |
| Wild-3 | *A. chukar* | Feathers | Contemporary | AQ | Duchessa | - | - |
| Wild-4 | *A. graeca* | Feathers | Contemporary | RI | Elefante | H3 | H3a |
| Wild-5 | *A. graeca* | Feathers | Contemporary | AQ | Orsello | H10 | H10d |
| Wild-6 | *A. graeca* | Feathers | Contemporary | RI | Pizzuto | H3 | H3b |
| Wild-7 | *A. graeca* | Feathers | Contemporary | AQ | Puzzillo | H8 | H8a |
| Wild-8 | *A. graeca* | Feathers-Feces | Contemporary | MC | Rotondo | H3 | 580 bp |
| Wild-9 | *A. graeca* | Feces | Contemporary | MC | Rotondo | H3 | 598 bp |
| Wild-10 | *A. graeca* | Feathers | Contemporary | MC | Rotondo | H3 | 588 bp |
| Wild-11 | *A. chukar* | Feathers | Contemporary | MC | Rotondo | - | - |
| Wild-12 | *A. chukar* | Feathers | Contemporary | MC | Rotondo | - | - |
| Wild-13 | *A. graeca* | Feces | Contemporary | MC | Rotondo | H10 | H10d |
| Wild-14 | *A. graeca* | Feathers | Contemporary | MC | Rotondo | H10 | 583 bp |

**Table 1.** *Cont.*

| Code | Species | Specimen | Dating | Province | Mountain/Area | 430 bp-H | 896 bp-H |
|---|---|---|---|---|---|---|---|
| Wild-15 | *A. graeca* | Feathers | Contemporary | MC | Rotondo | H10 | H10d |
| Wild-16 | *A. graeca* | Feathers | Contemporary | MC | Rotondo | H10 | H10d |
| Wild-17 | *A. graeca* | Feathers-Feces | Contemporary | AP | Sibilla | H8 | 560 bp |
| Wild-18 | *A. graeca* | Feathers | Contemporary | AP | Sibilla | H8 | 582 bp |
| Wild-19 | *A. graeca* | Feathers | Contemporary | AP | Sibilla | H3 | 582 bp |
| Wild-20 | *A. graeca* | Feathers | Contemporary | AP | Sibilla | H3 | 581 bp |
| Wild-21 | *A. chukar* | Feathers | Contemporary | FR | Ernici | - | - |
| Wild-22 | *A. graeca* | Feathers | Contemporary | AQ | Freddo | H24 | H24a |
| Wild-23 | *A. graeca* | Feathers | Contemporary | AQ | Ocre-Cagno | H8 | H8a |
| Wild-24 | *A. graeca* | Feathers | Contemporary | AQ | Ocre-Cagno | H3 | H3a |
| Wild-25 | *A. graeca* | Feathers | Contemporary | AQ | Freddo | H3 | H3a |
| Wild-26 | *A. graeca* | Feathers | Contemporary | MC | Rotondo | H3 | H3a |
| Wild-27 | *A. graeca* | Feathers | Contemporary | AQ | Greco-Maiella | H8 | H8a |
| Wild-28 | *A. graeca* | Feathers | Contemporary | AQ | Greco-Maiella | H8 | H8a |
| Wild-29 | *A. graeca* | Feathers | Contemporary | AQ | Greco-Maiella | H3 | H3a |
| Wild-30 | *A. graeca* | Feathers | Contemporary | AQ | Greco-Maiella | H3 | H3a |
| Wild-31 | *A. graeca* | Feathers | Contemporary | AQ | Greco-Maiella | H8 | H8a |
| Wild-32 | *A. graeca* | Feathers | Contemporary | AQ | Greco-Maiella | H3 | H3a |
| Wild-33 | *A. graeca* | Feathers | Contemporary | AQ | Greco-Maiella | H8 | H8a |
| Wild-34 | *A. graeca* | Feathers | Contemporary | PG | Aspra | H10 | H10d |
| Wild-35 | *A. graeca* | Feathers | Late 1990s | MC | Cardosa | H3 | H3b |
| Wild-36 | *A. graeca* | Feathers | Late 1990s | MC | Cardosa | H3 | 597 bp |
| Wild-37 | *A. graeca* | Feathers | Late 1990s | MC | Cardosa | H3 | H3a |
| Wild-38 | *A. graeca* | Feathers | Contemporary | AQ | Cornacchia | H3 | H3a |
| Wild-39 | *A. graeca* | Feathers | Contemporary | MC | Fema | H3 | n.d. |
| Wild-40 | *A. graeca* | Feathers | Contemporary | MC | Fema | H3 | n.d. |
| Wild-41 | *A. graeca* | Feathers | Late 1990s | MC | Fema | H3 | H3a |
| Wild-42 | *A. graeca* | Feathers | Late 1990s | MC | Fema | H10 | H10a |
| Wild-43 | *A. graeca* | Feathers | Late 1990s | MC | Fema | H3 | H3c |
| Wild-44 | *A. graeca* | Feathers | Late 1990s | MC | Fema | H3 | 558 bp |
| Wild-45 | *A. graeca* | Feathers | Late 1990s | MC | Fema | H3 | H3b |
| Wild-46 | *A. graeca* | Feathers | Contemporary | FR | Ernici | H3 | H3a |
| Wild-47 | *A. graeca* | Feathers | Contemporary | FR | Ernici | H3 | H3a |
| Wild-48 | *A. graeca* | Feathers | Contemporary | AQ | Freddo | H8 | H8a |
| Wild-49 | *A. chukar* | Feathers | Contemporary | AQ | Greco | - | - |
| Wild-50 | *A. graeca* | Feathers | Contemporary | IS | Matese | H24 | H24a |
| Wild-51 | *A. graeca* | Feathers | Contemporary | AQ | Orsello | H10 | H10d |
| Wild-52 | *A. graeca* | Feces | Contemporary | MC | Rotondo | H3 | H3a |
| Wild-53 | *A. graeca* | Feces | Contemporary | MC | Rotondo | H3 | 558 bp |
| Wild-54 | *A. graeca* | Feces | Contemporary | AP | Sibilla | H3 | 609 bp |
| Wild-55 | *A. graeca* | Feces | Contemporary | AP | Sibilla | H3 | 558 bp |
| Wild-56 | *A. graeca* | Feces | Contemporary | AP | Sibilla | H3 | 558 bp |
| Wild-57 | *A. graeca* | Feces | Contemporary | AP | Sibilla | H3 | 568 bp |
| Wild-58 | *A. graeca* | Feathers | Contemporary | AP | Sibilla | H3 | 605 bp |
| Wild-59 | *A. graeca* | Feathers | Contemporary | MC | Rotondo | H3 | H3a |
| Wild-60 | *A. graeca* | Feathers | Contemporary | MC | Rotondo | H3 | H3a |
| Wild-61 | *A. graeca* | Feathers | Contemporary | MC | Rotondo | H3 | H3a |
| Wild-62 | *A. graeca* | Feathers | Contemporary | MC | Rotondo | H3 | H3a |
| Wild-63 | *A. graeca* | Feathers | Contemporary | MC | Rotondo | H3 | H3a |
| Wild-64 | *A. graeca* | Feathers | Contemporary | MC | Rotondo | H3 | H3a |
| Wild-65 | *A. chukar* | Feathers | Contemporary | MC | Rotondo | - | - |
| Wild-66 | *A. chukar* | Feathers | Contemporary | MC | Rotondo | - | - |
| Wild-67 | *A. graeca* | Feathers | Contemporary | RI | Terminillo | H3 | H3a |
| Wild-68 | *A. graeca* | Feces | Contemporary | PG | Ventolosa | H3 | H3a |
| Wild-69 | *A. graeca* | Feathers | Contemporary | AQ | Greco | H3 | H3a |

Five haplotypes were previously found throughout the Apennines by Randi et al. [2]: H3, H8, H12, H23 and H24, with frequencies equal to 3% (H12, H23 and H24), 41% (H3) and 51% (H8). During the present survey, haplotypes H3, H8 and H24 were confirmed in the Apennines with frequencies of 67.2% (n° 41 from 10 sampling areas), 16.4% (n° 10 from six sampling areas) and 3.3% (n° 2 from two sampling areas), respectively. Although the H10 haplotype was not previously registered for the Apennine region, in the present survey, it was found with a frequency of 13.1% (n° 8) in five independent sampling areas (Orsello Mountain—AQ, Rotondo Mountain—MC, Aspra mountain—PG and Fema Mountain—MC). Furthermore, it matches to a taxidermized specimen from Fema Mountain (MC) dated to the late 1990s, assessing a long-standing presence of this haplotype in the Central Apennine. Other haplotypes previously recorded in the Apennines differ from H10 only for one nucleotide, as for example C/T at position 245 for H3 or for only one deletion at position 191 for H23. Our data should be pooled to the previous ones [4], providing an extended and more reliable list of the haplotypes conserved in the wild extant Apennine populations. This new insight, even when limited to the 430 bp sequences, can immediately support the conservation purposes targeted by the National Action Plan [5]. Indeed, the knowledge of the haplotypes conserved in wild extant populations allows for the identification of breeding farm that have eligible stocks for supporting reintroduction and restocking actions, and in the case of unavailability of suitable captive stocks, it is advisable to start ex situ conservation programs sourcing founders of adequate genotypes from natural populations. Furthermore, analyzing the pooled dataset searching for phylogenetic patterns could promote a better understanding of the degree of genetic diversification of the Apennine population [4].

Thanks to the extended fragments (896 bp), a total of 12 long sequences (896 bp), corresponding to 7 new haplotypes from the present survey focused on the Apennine population (Table 1) and other 6 found for specimens of Alpine or dubious origins, were identified and registered in GenBank with access codes from ON086317 to ON086329 (Table 2).

**Table 2.** List of the 12 new haplotypes of *Alectoris graeca* identified on the base of the mutations recorded in the tract between 431 and 896 bp of the long sequences.

| Haplotypes (896 bp) | GenBank Codes | Sampling Locations | Haplotype (430 bp) | Variable Sites | | | | |
|---|---|---|---|---|---|---|---|---|
| | | | | 4 3 4 | 6 2 9 | 8 0 1 | 8 2 0 | 8 5 1 |
| H3a-long | ON086324 | Apennine | H3+ | T | T | A | C | T |
| H3b-long | ON086325 | Apennine | H3+ | T | T | T | C | T |
| H3c-long | ON086326 | Apennine | H3+ | G | T | A | C | T |
| H3d-long | ON086327 | Apennine | H3+ | T | A | A | C | T |
| H6a-long | ON086328 | Alps | H6+ | T | T | A | C | T |
| H8a-long | ON086329 | Apennine | H8+ | T | T | A | C | T |
| H10a-long | ON086317 | Apennine | H10+ | T | T | T | A | T |
| H10b-long | ON086318 | Apennine | H10+ | T | T | T | C | T |
| H10c-long | ON086319 | Greece | H10+ | T | T | A | C | A |
| H10d-long | ON086320 | Apennine | H10+ | T | T | A | C | T |
| H24a-long | ON086322 | Apennine | H24+ | T | T | A | C | T |
| H29a-long | ON086323 | Macedonia | H29+ | T | T | A | C | T |

In Table 2, for each new 896 bp haplotype (column 1), the GenBank access code (column 2), the sampling locations, the corresponding 430 bp haplotype [4] (column 4) and the characteristic variable sites (columns from 5 to 10) were provided.

In total, 4 haplotypes of the 12 long new (896 bp) share the 430 bp sequence typical of H3 (from H3a-long to H3d-long), the same number share the 430 bp sequence typical

of H10 (from H10a-long to H10d-long), while the others five haplotypes (H6, H8, H13, H24 and H29) did not show polymorphisms along the fragment; therefore, only one long sequence each was found and named according to the indications above (Table 2).

Among these, 7 new long haplotypes (896 bp) were confirmed in the Apennines (access codes: ON086317, ON086320, ON086322, ON086324, ON086325, ON086326, ON086329), three referable to H3 (H3a-long, H3b-long and H3c-long), two to H10 (H10a-long and H10d-long) and one each to H8 (H8a-long) and H24 (H24a-long) (Table 2), with different frequencies (Table 1).

The improvement of the fragment length and the identification of new haplotypes on the base of mutations in the tract between 431 and 896 bp might be a finer tool allowing a more detailed evaluation of the biological diversity at the individuals level potentially favoring (a) the correct selection of founders preserving local adaptations for ex situ and in situ conservation purposes and (b) a better discrimination between disjoint populations. This insight can be useful at both intra-population level, for highlighting the eventual loss of connectivity and consequent sub-population isolation, and inter-population level, for assessing the degree of genetic differentiation among the Alps, Balkans and Apennines. At this scope, we call for sharing long sequences (896 bp) from different crucial areas (i.e., Dinaric Alps and Albania) [4] of the European distribution range.

**Author Contributions:** Conceptualization: P.V., A.A. and L.L.; Methodology: L.B., P.V., L.L., P.G., A.R. and S.C.; Writing—original draft preparation: L.B., P.V. and L.L.; Writing—review and editing: P.V., S.C., L.E., R.P., A.A. and L.B.; Funding acquisition: L.L. and A.A.; Supervision: L.L. and A.A. All authors have read and agreed to the published version of the manuscript.

**Funding:** This research was supported by MIUR initiative "Department of excellence" (Law 232/2016) granted to DAFNE, University of Tuscia (Italy), by Sibillini Mountains National Park and Italian Minister for Ecological Transition (MITE) and by "Progetto Ricerca Di Base 2020", University of Perugia (Italy).

**Institutional Review Board Statement:** The study was conducted in accordance with the European rules (Directive 2010/63/UE). The project was approved by the ethical committee of Perugia University (prot. 7/2022).

**Informed Consent Statement:** Not applicable.

**Data Availability Statement:** Data are available at the link: https://www.ncbi.nlm.nih.gov/ (GenBank accession numbers were provided on 28 March 2022) with the following codes: ON086324, ON086325, ON086326, ON086327, ON086328, ON086329, ON086317, ON086318, ON086319, ON086320, ON086322, ON086323.

**Acknowledgments:** The authors thank the museum technicians and the private collectors for their help and cooperation. Furthermore, the authors would like to thank Stefano Ripert and Dario Bondani (University of Tuscia, VT, Italy) for the support in field sampling, and Federico Morandi (Sibillini Mountains National Park, Visso, MC, Italy) for the technical support.

**Conflicts of Interest:** The authors declare no conflict of interest. The funders had no role in the design of the study; in the collection, analyses or interpretation of data; in the writing of the manuscript, or in the decision to publish the results.

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
