# Peer review of "An Updated List of Rock Partridge (Alectoris graeca) Haplotypes from the Apennines—Central Italy"

_data, 2022_

Round 1

Reviewer 1 Report

In the §1 SUMMARY, lines 46-4, you should recall that in Italy, Sicilian rock partridge (RP) populations face the same conservation problems (isolation and fragmentation) than Apennines RP  populations.Cf :

https://webgate.ec.europa.eu/life/publicWebsite/project/details/3245  

You also should mention the historical background about uncertain taxonomical status of Apennine rock partridges. Some authors considered them as a subspecies (cf. PRIOLO A., 1984 - Variabilità in Alectoris graeca e descrizione di A. graeca orlandoi subsp. nova degli Appennini.Riv. ital. Orn., 54: 45-76.).

Lines 51-54:  recall that, before the publication of Randi et al. 2003, the first phylogenetic tree of Rock partridge (RP) populations showing the affinity between samples from Apennine RP populations and Albanian RP samples was established since 1998 : read pages 410-411 in Randi E. Lucchini V. & Bernard-Laurent A. 2008 . Evolutionary genetics of the Alectoris partridges : the generation and conservation of genetic diversity at different time and space scales. Game Wildl; vol.15 (Hors série): 407-415.

In the §2.2, it would be useful to provide some informations about presence of wild chukar in the Apennines because the rate of samples identified as chukar is rather high (12%) (i.e. ageing of the last releases of chukar partridges in the Apennines, areas of Apennines where chukar releases were undertaken ?)

Dataset presented in §3 is technically sound. It would be advisable to mention potential methodological difficulties of genotyping samples from feathers in comparison with faeces for future similar research. Was genotyping as efficient, using feathers or faeces ?

Data have been archived appropriately and are available for reuse. 

Reviewer 2 Report

GENERAL COMMENT:

I consider this work is within the scope of “Data”. It contains information useful in a field in which available information is scarce and of interest to improve knowledge on Rock partridge (Alectoris graeca) genetics and conservation in the wild. Overall, it is well organised. However, I indicate several improvements to be performed in the manuscript.

ABSTRACT:

Line 16: Type “Alectoris graeca” in italics.

SUMMARY:

Lines 33-34: I suggest adding a sentence with a brief morphological description of the Rock partridge.

Lines 33-34: I suggest adding a brief, but more specific, indication of the distribution of this species. It has been indicated only that is “endemic to Europe”.

Near Line 57: I suggest formulating more clearly the objective of this manuscript. For example, by modifying sentence in Lines 57-60 in a way similar to: “This work aims to report an updated and expanded dataset of Rock Partridge haplotypes that improves the size and the distribution of the Apennine samples allowing the identification of previously undetected haplotypes and the description of new ones based on mtDNA fragments longer than the previously sequenced”.

METHODS:

Lines 66-68: Please indicate that the samples were from Rock partridges (Alectoris graeca).

DATA DESCRIPTION:

Line 105: Insert space between words at: “(column seven)and”, thus resulting in: “(column seven) and”.

Line 145: Add space between table and text paragraph.

REFERENCES SECTION:

In general terms, this section is well organised and adjusted to the style of the journal for references. However, some improvement is possible. For example:

Line 195: Type “Quad. Cons.” With the full words.

Line 198: Remove one of the duplicated “ISBN” at: “ISBN ISBN”.

Line 227: Type “italica” in italics.

In many references, for homogeneity: Do not capitalize the first letter of all words in article titles (except proper names)

TABLES:

Tables need to be interpreted independently of the manuscript text. Therefore, some improvement is needed:

Table 1: Title: Please indicate that it refers to Rock partridge (Alectoris graeca).

Table 1: Write “Contemporary rather than “Contemp.” From Wild-47 row.

Table 1: Write the provinces with the full words rather than initials (“Preugia” rather than “PG”, etc.)

Table 2: Title: Please indicate that it refers to Rock partridge (Alectoris graeca).
